# Identification and Characterization of a Novel Hypovirus from the Phytopathogenic Fungus *Botryosphaeria dothidea*

**DOI:** 10.3390/v15102059

**Published:** 2023-10-07

**Authors:** Yongqi Wen, Jinyue Qu, Honglin Zhang, Yi Yang, Rui Huang, Jili Deng, Jiayu Zhang, Yanping Xiao, Jiali Li, Meixin Zhang, Guoping Wang, Lifeng Zhai

**Affiliations:** 1College of Life Science and Biotechnology, Yangtze Normal University, Chongqing 408100, China; lay18716418559@163.com (Y.W.); m15095891074@163.com (J.Q.); tammymills452@gmail.com (H.Z.); m18985049426@163.com (Y.Y.); hr15223119451@sina.cn (R.H.); dengd2023@outlook.com (J.D.); 15025580561@163.com (J.Z.); xyp010125@163.com (Y.X.); ljl01122737@163.com (J.L.); zhangmeixin00@126.com (M.Z.); 2Key Lab of Plant Pathology of Hubei Province, College of Plant Science and Technology, Huazhong Agricultural University, Wuhan 430070, China; gpwang@mail.hzau.edu.cn

**Keywords:** *Botryosphaeria dothidea*, Botryosphaeria dothidea hypovirus 1, genome, *Betahypovirus*, *Hypoviridae*

## Abstract

Many mycoviruses have been accurately and successfully identified in plant pathogenic fungus *Botryosphaeria dothidea*. This study discovered three mycoviruses from a *B*. *dothidea* strain SXD111 using high-throughput sequencing technology. A novel hypovirus was tentatively named Botryosphaeria dothidea hypovirus 1 (BdHV1/SXD111). The other two were known viruses, which we named Botryosphaeria dothidea polymycovirus 1 strain SXD111 (BdPmV1/SXD111) and Botryosphaeria dothidea partitivirus 1 strain SXD111 (BdPV1/SXD111). The genome of BdHV1/SXD111 is 11,128 nucleotides long, excluding the poly (A) tail. A papain-like cysteine protease (Pro), a UDP-glucose/sterol glucosyltransferase (UGT), an RNA-dependent RNA polyprotein (RdRp), and a helicase (Hel) were detected in the polyprotein of BdHV1/SXD111. Phylogenetic analysis showed that BdHV1/SXD111 was clustered with betahypovirus and separated from members of the other genera in the family *Hypoviridae*. The BdPmV1/SXD111 genome comprised five dsRNA segments with 2396, 2232, 1967, 1131, and 1060 bp lengths. Additionally, BdPV1/SXD111 harbored three dsRNA segments with 1823, 1623, and 557 bp lengths. Furthermore, the smallest dsRNA was a novel satellite component of BdPV1/SXD111. BdHV1/SXD111 could be transmitted through conidia and hyphae contact, whereas it likely has no apparent impact on the morphologies and virulence of the host fungus. Thus, this study is the first report of a betahypovirus isolated from the fungus *B*. *dothidea*. Importantly, our results significantly enhance the diversity of the *B*. *dothidea* viruses.

## 1. Introduction

Mycoviruses are widely distributed and reported in the major taxonomic groups of fungi [1,2,3,4]. The majority of known mycoviruses with double-stranded RNA (dsRNA), positive-sense single-stranded RNA (+ssRNA), negative-sense single-stranded RNA (-ssRNA), and single-stranded circular DNA (ssDNA) genomes have been reported [1,2]. The *Hypoviridae* is a viral family that contains capsidless + ssRNA viruses [5]. This family’s 7.3–18.3 kb genomes encode either a single large ORF or two ORFs [5]. Additionally, hypoviruses have been detected in many filamentous fungi and invertebrates [5,6,7]. Notably, some hypoviruses induce hypovirulence to host fungi, while others do not [2,4]. Currently, the family *Hypoviridae* has eight genera, including *Alphahypovirus*, *Betahypovirus*, *Gammahypovirus*, *Deltahypovirus*, *Epsilonhypovirus*, *Zetahypovirus*, *Thetahypovirus* and *Etahypovirus* [5].

*Botryosphaeria dothidea*, a widespread plant pathogenic fungus, can generally cause leaf spot, stem canker, stem wart, and fruit rot diseases on many economically important plants [8]. This pathogen infects many important fruit trees in China, such as apple, grapevine, kiwifruit, peach, and pear [9,10,11,12,13,14,15]. In recent decades, many mycoviruses have been isolated and characterized from this plant pathogenic fungus. Notably, the dsRNA viruses isolated from *B*. *dothidea* belong to the families *Chrysoviridae*, *Partitiviridae*, *Totiviridae*, *Botybirnaviridae*, and *Polymycoviridae* [16,17,18,19,20,21,22,23,24]. Recent studies have also reported nine positive ssRNA viruses infecting this fungus. Among them, Botryosphaeria dothidea fusarivirus 1 (BdFV1) and BdFV2 belong to the family *Fusariviridae* [25,26]. Botryosphaeria dothidea botourmiavirus 1 (BdBOV-1), Botryosphaeria dothidea ourmia-like virus 1 (BdOLV-1), and BdOLV-2 cluster into the family *Botourmiaviridae* [27,28,29]. Botryosphaeria dothidea botrexvirus 1 (BdBV1) is member of a newly proposed family, *Alphaflexiviridae* [30]. In addition, three mitoviruses named Botryosphaeria dothidea mitovirus 1 (BdMV1), BdMV2, and BdMV3 are classified in the family *Mitoviridae* [31,32,33]. Recently, six viroid-like single-stranded circular RNAs, ranging in size from 157 to 450 nucleotides, were isolated from the fungus *B*. *dothidea* [34]. Therefore, mycoviruses infected *B*. *dothidea* are widespread and exhibit high diversity.

Some mycoviruses can cause host phenotypic changes, such as abnormal growth, morphological changes, and hypovirulence [4]. It is worth noting that hypovirulence-associated mycoviruses can potentially prevent and treat plant fungal diseases [4]. Currently, four viruses in *B*. *dothidea* strains lead to hypovirulence in their host, such as Botryosphaeria dothidea chrysovirus 1 (BdCV1), Botryosphaeria dothidea botybirnavirus 1 (BdBRV1), Botryosphaeria dothidea polymycovirus 1 (BdPmV1), and BdBV1 [16,19,23,24,30].

In this study, three mycoviruses, including a novel hypovirus, BdPmV1, and Botryosphaeria dothidea partitivirus 1 (BdPV1), were detected from the strain SXD111 of *B*. *dothidea*. The novel hypovirus was tentatively named Botryosphaeria dothidea hypovirus 1 (BdHV1/SXD111). The genomic structure and molecular evolution analysis indicated that BdHV1/SXD111 is a new member of the genus *Betahypovirus* in the family *Hypoviridae*. Moreover, we obtained the full cDNA sequences of the other two known viruses, which we named BdPmV1/SXD111 and BdPV1/SXD111. Interestingly, the smallest dsRNA segment in the strain SXD111 was a novel satellite component for the helper virus BdPV1/SXD111.

## 2. Materials and Methods

### 2.1. Fungal Strains and Culture Conditions

Seven strains of *B*. *dothidea* were used in this study (Table 1). Strain SXD111 was isolated from a stem wart sample on a pear tree (*Pyrus bretschneideri* cv. ‘Hongxiangsu’) growing in Shaanxi Province, China [10]. The strain YZN115 infected by BdPmV1 [23] and strain YCLYY11 coinfected by BdCV1, BdPV1, and Botryosphaeria dothidea victorivirus 3 (BdVV3) [17] were used as the positive control in high-throughput sequencing. Strains JST1, JST2, and JST3 were obtained from the horizontal transmission assay (Table 1). The virus-free strain JNT1111 was used as a control to evaluate biological characteristics [10]. All strains were cultured on potato dextrose agar (PDA) plates at 28 °C and stored on PDA slants at 4–6 °C.

### 2.2. RNA Extraction, High-Throughput Sequencing, and Data Analysis

Mycelia of strains SXD111, YZN115, and YCLYY11 were cultured on cellophane membranes laid on PDA plates (90 mm in diameter) for five days at 28 °C. Fresh mycelia were harvested and ground into a fine powder in liquid nitrogen. The total RNA was extracted from fresh mycelia using a TRIzol RNA extraction kit (Thermo Fisher Scientific, Waltham, MA, USA). The extracted RNA samples were detected using agarose gel electrophoresis with 1.0% (*w*/*v*) and assessed using Nanodrop (Thermo Fisher Scientific, Madison WI, USA). Then, qualified RNA samples of tested strains were mixed in equal amounts. The mixed RNA sample was sent to the Novogene Bioinformatic Technology Company (Beijing, China) and constructed for an rRNA-depleted RNA-seq cDNA library [35]. In short, rRNA in the mixed RNA sample was removed, and then double-stranded cDNAs were synthesized using random hexamers primer (N6). Subsequently, the library was obtained using the polymerase chain reaction (PCR). Finally, the quality of the library was tested and sequenced using an Illumina HiSeq. The raw output reads were processed to remove adaptors and poor-quality reads. Furthermore, the remaining clean reads were assembled. Moreover, contigs were annotated using a local BLASTx search restricted to the mycoviral sequences, which were retrieved from the National Center for Biotechnology Information (NCBI) database.

### 2.3. Extraction and Purification of dsRNA

The extraction of dsRNA from strains SXD111, YZN115, YCLYY11, and JNT1111 was performed using the cellulose chromatography (Sigma–Aldrich, Dorset, England) method [36]. The obtained dsRNA samples were treated using DNase I and S1 nuclease (TaKaRa, Dalian, China) following the manufacturer’s instructions. Furthermore, digested dsRNA elements were separated in 1.0% agarose gel at 100 V and visualized in a UV transilluminator after staining with 0.1% (*v*/*v*) 4S green plus nucleic acid (Sangon, Shanghai, China).

### 2.4. Validation of Virus-like Contigs via RT-PCR and Viral Sequencing Amplification

We used reverse-transcription polymerase chain reaction (RT-PCR) to determine mycoviruses in the tested strains using specific primers that were designed based on the assembled contigs (Appendix A). The total RNA (1.0 µg) was mixed with 0.01 µg of a random hexamer primer (TaKaRa, Dalian, China) for cDNA synthesis. Next, according to the manufacturer’s instructions, Moloney murine leukemia virus (M-MLV) reverse transcriptase (Simgen, Hangzhou, China) was added into the reaction. After a reaction for 1 h at 37 °C, obtained products were used for PCR or stored at −20 °C.

The full-length sequence of each virus was obtained using RT-PCR and the rapid amplification of cDNA ends (RACE) protocol. Due to a lack of useful contig, the cDNA sequence of the smallest dsRNA was determined following a reported method [19]. All primers used in these assays are listed in Appendix A. The PCR products were electrophoresed in 1.2% agarose gel for visualization on a UV trans-illuminator. We also used a pMD18-T vector (TaKaRa, Dalian, China) to ligate the purified PCR product, and then the ligated product was transformed into competent cells of *Escherichia coli* DH5α. Furthermore, sequencing was performed by Beijing Tsingke Biotech Co., Ltd. (Beijing, China). At least three independent clones of each product were detected in both orientations.

The terminal sequences of the dsRNA virus were obtained using the RNA ligase-mediated rapid amplification of the cDNA ends (RLM-RACE) procedure previously described [37]. We also amplified the cDNA ends of the ssRNA virus using a SMARTer RACE 5′/3′ kit (TaKaRa, Dalian, China) according to the manufacturer’s instructions. All purified PCR products were ligated into a pMD18-T vector and transformed into *E*. *coli* DH5α competent cells. At least five independent clones of each product of cDNA ends were sequenced.

### 2.5. Sequences Alignment and Phylogenetic Analysis

The cDNA sequences of dsRNAs were assembled using the DNAMAN software version 8 (Lynnon Corporation, Pointe-Claire, QC, Canada). Sequence searches were conducted using the BLASTx or BLASTp program on the NCBI website (https://blast.ncbi.nlm.nih.gov/Blast.cgi accessed on 21 August 2023). The open reading frames (ORFs) of obtained sequences were predicted using the ORFfinder website (https://www.ncbi.nlm.nih.gov/orffinder/, accessed on 21 August 2023). Conserved domains in the protein sequence were identified using a conserved domain search service (https://www.ncbi.nlm.nih.gov/Structure/cdd/wrpsb.cgi accessed on 21 August 2023) [38]. Multiple sequence alignments were performed using MAFFT software version 7 (https://mafft.cbrc.jp/alignment/server/, accessed on 21 August 2023) [39]. The aligned conserved amino acid sequences were visualized using the GeneDoc software version 2.7. Furthermore, phylogenetic tree was constructed using maximum likelihood and MEGA X software [40]. Bootstrap value was evaluated with 1000 replicates.

### 2.6. Vertical and Horizontal Transmission of the Viruses

The conidia of strain SXD111 were induced on the PDA plate under alternating cycles of light/dark (16 h:8 h) and near-UV light (365 nm) as previously described [10]. Conidia were individually picked out and transferred into fresh PDA plates. The viruses in single conidia (SC) strains were detected via RT-PCR using specific primers (Appendix A). A horizontal transmission assay was executed on PDA in Petri dishes using the hyphal contact technique as described previously [23]. The strain SXD111 was the donor, and the virus-free strain JNT1111 served as the recipient. Each strain was repeated with three Petri dishes. The dual cultures were incubated for 10 days at 28 °C. Then, the derivative strains were obtained from the colonial margin of strain JNT1111 (the farthest site from the contact zone of strains SXD111 and JNT1111). The derivative strains were named SJT1, SJT2 and SJT3. Furthermore, the presence of the viruses in derivative strains SJT1, SJT2 and SJT3 were determined via RT-PCR using specific primers (Appendix A). Moreover, the colony morphology, growth rates, and virulence of these derivatives were evaluated and compared with those of their parental strains SXD111 and JNT1111.

### 2.7. Biological Properties of Fungal Strains

The growth rates and morphology of virus-infected and free strains of *B*. *dothidea* were estimated (Table 1). All freshly grown mycelial agar plugs (5 mm in diameter) were cultured on a PDA plate (90 mm in diameter) in quintuplicate at 28 °C in the dark for five days. The colony morphology and mycelial diameter were estimated every 24 h. The virulence of each strain was assessed on detached pear fruits (*P*. *brettschneider* cv. “Huang guan”) according to a previously described method [10]. All pear fruits were selected and shared a consistent appearance and texture. Fungal strains were then inoculated on wounded mature pear fruits using colonized agar plugs (5 mm in diameter), and vaccination sites were covered with sterilized moist cotton balls to maintain humidity. The incubated fruits were placed in closed plastic baskets and deposited in an illumination incubator at 28 °C. Each strain of inoculation was repeated with four fruits. Moreover, the diameters of the fruit lesions were recorded daily for up to four days post-inoculation (dpi). Each of the above tests had two replicates.

## 3. Results

### 3.1. Diversity of Viruses in the Strains SXD111, YZN115, and YCLYY11

To identify the presence of mycoviruses in the strain SXD111 of the *B*. *dothidea*, an RNA sequencing library with strains SXD111, YZN115, and YCLYY11 was generated. The library was sequenced and assembled at a considerable depth. After removing rRNA from the assembled data, we obtained 24,092,408 reads via fragment reverse transcription and screened out 24,092,344 clean reads using quality control data. Overall, a total of 15,053 contigs totaling 12,418,301 bp were obtained by transcriptional splicing, database alignment, and coding region prediction. The N50 and N90 values were 1033 and 395, respectively. A total of 15 contigs derived from mycoviruses were retrieved using a local BLASTx search (Table 2).

BLASTx analysis revealed that the predicted amino acid sequence of contig 1 (11,113 nt) best resembled the polyprotein of Cryphonectria hypovirus 4 (CHV4) and had a 49.77% homology (Table 1). Thus, the sequence of contig 1 resulted from a novel hypovirus in the family *Hypoviridae*, which we tentatively named Botryosphaeria dothidea hypovirus 1 (BdHV1). In addition, sequences of four known viruses that infected the tested strains were also discovered from the other contigs (Table 2).

### 3.2. Detection and Validation of Mycoviruses in the Test Strains

DsRNAs were extracted from the *B*. *dothidea* strain SXD111 and subjected to digestion with DNase I or the S1 nuclease. Agarose gel electrophoresis revealed that at least eight dsRNA fragments ranging from 0.5 to 10.0 kbp were obtained in this strain (Figure 1A). DsRNA segments of BdPmV1, BdPV1, and BdCV1 were extracted from strains YZN115 and YCLYY11 (Figure 1B). Moreover, dsRNA bands similar in size to BdPmV1 and BdPV1 were also found in the strain SXD111 (Figure 1B). We concluded that the dsRNA bands of 0.5 to 2.5 kbp in strain SXD111 were segments of BdPmV1 and BdPV1, respectively (Figure 1A). These results were also confirmed using RT-PCR with primers specific to segments of BdHV1, BdPmV1, BdPV1, BdCV1, and BdVV3 (Figure 1C). Thus, these data strongly support that strain SXD111 was coinfected with three viruses: a novel hypovirus BdHV1/SXD111, a polymycovirus BdPmV1/SXD111, and a partitivirus BdPV1/SXD111 (Figure 1).

### 3.3. Genetic Analysis of BdHV1/SXD111

The full-length cDNA sequence of BdHV1/SXD111 was obtained using RT-PCR and RACE cloning. Notably, sequence analysis showed that the total genome length of BdHV1/SXD111 was 11,128 nt, excluding poly(A), with a 50.3% GC content. ORFfinder prediction revealed that the genetic sequence of BdHV1/SXD111 contained one large ORF from nucleotide 914 to 10,687 nt, encoding a protein (P1) of 3257 amino acids with a predicted molecular mass of 369 kDa (Figure 2A). Furthermore, the length of 5’- and 3’-untranslated regions of BdHV1/SXD111 were 913 and 441 nt, respectively (Figure 2A). The complete sequence of BdHV1/SXD111 was assembled and deposited in the GenBank database (accession number OR387868).

A BLASTp search using the deduced amino acid sequence of P1 in NCBI showed that the protein had sequence identities with the polyprotein encoded by Cryphonectria hypovirus 4 (CHV4, E-value = 0; identity = 49.77%), Alternaria dianthicola hypovirus 1 (AdHV1, E-value = 0; identity = 49.70%), and other betahypoviruses (Appendix A). Thus, the protein P1 decoded by BdHV1/SXD111 was a polyprotein of the novel hypovirus. The motif search for the amino acid sequence showed that three conserved domains were found in the polyprotein of BdHV1/SXD111, including a UDP-glucose/sterol glucosyltransferase (UGT; Pfam 340817, E value = 2.72 × 10^−7^), an RNA dependent RNA polymerase (RdRp; Pfam00680, E value =1.90 × 10^−10^), and an RNA helicase (Hel; Pfam 00271, E value = 0.0068) (Figure 2A).

Multiple sequence alignment based on the amino acid sequence of RdRp domains of BdHV1/SXD111 and other hypoviruses revealed nine conserved motifs in these hypoviruses, including the S/GDD motif, which is a typical feature in the RdRp of hypoviruses (Figure 2B). Amino acid identities of RdRp domains between BdHV1/SXD111 and other betahypoviruses were 66.23% to 74.41% (Appendix A). Multiple alignment also showed that seven conserved motifs were conserved among helicase domains of BdHV1/SXD111 and other hypoviruses (Figure 2C). Amino acid identities of helicase domains between BdHV1/SXD111 and other betahypoviruses were 49.12% to 61.21% (Appendix A). Multiple alignment also showed that four amino acid regions were conserved among UGT domains of BdHV1, CHV3, CHV4, and other betahypoviruses (Figure 2D). Amino acid identities of UGT domains between BdHV1/SXD111 and other betahypoviruses ranged from 52.96% to 58.79% (Appendix A). Although conserved motifs of papain-like protease (Pro) in the polyprotein of BdHV1/SXD111 were not found by the motif search and CD search in NCBI, the result of the alignment of the N-terminal region of the polyproteins of BdHV1/SXD111 and some hpoviruses revealed that the conserved cysteine, histidine, and glycine core residues were found in the polyprotein of BdHV1/SXD111 (224−302 aa region) (Figure 2E). This result indicated that the N-terminal region of the polyprotein encoded by BdHV1/SXD111 contained a Pro domain (Figure 2A,E).

### 3.4. Phylogenetic Analysis of BdHV1/SXD111

The phylogenetic tree was constructed using the polyproteins of BdHV1/SXD111 and reported members of the family *Hypoviridae*. Phylogenetic analysis showed that BdHV1/SXD111, CHV4, Fusarium concentricum hypovirus 1 (FcHV1), Fusarium oxysporum dianthi hypovirus 2 (FodHV2), Setosphaeria turcica hypovirus 1 (StHV1), and other batahypoviruses were closely related and clustered in the same clade to form a clade (Figure 3). In summary, the genomic structure and molecular evolution analysis indicate that BdHV1/SXD111 is a new member of the genus *Betahypovirus* in the family *Hypoviridae*.

### 3.5. Genetic Analysis of BdPmV1/SXD111 and BdPV1/SXD111

We also obtained the cDNA sequence of the BdPmV1/SXD111 and BdPV1/SXD111 segments by combining RT-PCR and RLM-RACE sequences. The full-length dsRNA1 to 5 of the BdPmV1/SXD111 were 2396 bp, 2232 bp, 1966 bp, 1131 bp, and 1060 bp, respectively. According to BLASTn, the sequences best resembled corresponding segments of the BdPmV1 strain YZN115 (BdPmV1/YZN115) and had 96.79%, 97.92%, 93.34%, 97.35%, and 97.17% homologies, respectively (GenBank: KP245734−KP245738, E-value = 0, coverage 97–100%). We deposited the corresponding sequences of BdPmV1/SXD111 in GenBank under accession number OR397571−OR397575.

The full-length dsRNAs of BdPV1/SXD111 were 1823 bp, 1623 bp, and 557 bp, respectively. The 5′-UTRs of dsRNA1 and dsRNA2 of the virus were 34 and 51 bp long (Figure 4A). Notably, a 23 bp conserved sequence (CGAAAAUGAGUCACAACAUUACA) was contained at their 5′-termini. The corresponding 3′-UTRs of dsRNA1 and dsRNA2 were 70 and 87 bp long (Figure 4A), respectively, while containing 11 bp conserved sequences (CCCUAACACCA). According to BLASTn, the sequences best resembled corresponding RdRp and coat protein (CP) genes of the BdPV1 strain LW-1 (BdPV1/LW-1) and had 97.53% and 98.46% homologies, respectively (GenBank: KF688740 and KF688741, E-value = 0).

The complete nucleotide sequence of BdPV1/SXD111-dsRNA3 was 557 bp in length and had a small putative ORF (nt positions 126–203) on its negative strand coding for a protein of 25 amino acid residues. Interestingly, the sequence of dsRNA3 of BdPV1/SXD111 shared no similarity with the sequences deposited in the NCBI database or satellite-like RNA of BdPV1/LW-1. The alignment of dsRNA3 with dsRNA1 and dsRNA2 of BdPV1/SXD111 indicated that 7 bp (CGAAAAU) at the 5′-termini and 6 bp (ACACCA) at the 3′-termini were 100% identical (Figure 4B). Additionally, the 5′- and 3′- termini of BdPV1/SXD111-dsRNA3 were predicted to be capable of forming stem-loop structures (Figure 4C). By contrast, the remaining sequence had no detectable similarity with both dsRNAs of BdPV1/SXD111. Therefore, dsRNA3 is a novel satellite-like RNA of BdPV1/SXD111. We deposited the corresponding sequences of BdPV1/SXD111 in GenBank under accession number OR387874−OR387876.

### 3.6. Vertical and Horizontal Transmission of BdHV1/SXD111

A total of 146 single-conidium (SC) strains were derived from SXD1111. The RT-PCR results revealed that all SC strains were positive for BdHV1/SXD111 and BdPmV1/SXD111. Also, BdPV1/SXD111 was negatively detected in 79 SC strains. These results revealed that BdHV1/SXD111 could vertically transmit in host strain. In horizontal transmission assays, strain JNT1111 was used as the recipient in the hyphal contact with strain SXD111. The result of RT-PCR for detecting BdHV1/SXD111, BdPmV1/SXD111, and BdPV1/SXD111 demonstrated that sub-strains SJT1, SJT2 and SJT3 harbored BdHV1/SXD111 (Figure 5A). However, BdPmV1/SXD111 and BdPV1/SXD111 were negatively detected in all sub-strains (Figure 5A).

### 3.7. Effects of BdHV1/SXD111 on B. dothidea

To obtain the virus-free BdHV1/SXD111 isogenic lineage from strain SXD111, a hyphal tip culture, single conidium, and protoplast regeneration methods were conducted. However, all methods failed. Meanwhile, to check the effects of BdHV1/SXD111 on the biological characteristics of *B*. *dothidea*, the colony morphology and growth rate of strains SXD111, JNT1111, SJT1, SJT2, and SJT3 were determined. The sub-strains SJT1, SJT2, and SJT3 were isogenic lineages of strain JNT1111. Compared with the strain JNT1111, the colonial morphology of sub-strains SJT1, SJT2, and SJT3 were not significantly different (Figure 5B). Additionally, the growth rates on PDA of sub-strains SJT1, SJT2, and SJT3 did not significantly differ from that of strain JNT1111 (27.5–28.9 mm/day vs. 27.0 mm/day) (Figure 5C). Moreover, the growth rates did not significantly differ between strains JNT1111 and SXD111 (Figure 5C). All tested strains induced similar-sized lesions when inoculated on wounded pear fruits (*P*. *bretschneideri* cv ‘Huangguan’) (Figure 5D,E). Particularly, the virulence of these sub-strains SJT1, SJT2, and SJT3 and their mother strain JNT1111 did not differ significantly (Figure 5D). Therefore, these results suggest that BdHV1/SXD111 confers no obvious effects on the morphologies and virulence of *B*. *dothidea*.

## 4. Discussion

To date, many *B*. *dothidea* strains have been reported to possess mycoviruses. However, no research on hypovirus showed it to infect any strains of this fungus. In this study, from a *B*. *dothidea* strain SXD111, we isolated and characterized a novel hypovirus coinfeted with BdPmV1/SXD111 and BdPV1/SXD111. Additionally, BdPV1/SXD111 harbored a novel satellite-like RNA.

Viruses are commonly present in *B*. *dothidea*. Currently, 18 mycoviruses in 10 families have been isolated from this fungus. In these viruses, we also identified BdPmV1 and BdPV1 from strain SXD111 in our study. Initially, BdPmV1 and BdPV1 were found from the different strains of *B*. *dothidea*, BdPmV1 isolated from strain YZN115 and BdPV1 isolated from strain LW-1 [19]. Notably, BdPmV1/SXD111 and BdPV1/SXD111 had a strong similarity to the sequences of BdPmV1/YZN11 and BdPV1/LW-1 with 93.34%−98.46% identities, respectively. Therefore, based on the genomic structure and molecular evolution analysis, BdPmV1/SXD111 and BdPV1/SXD111 isolated from strain SXD111 were new strains of viruses BdPmV1 and BdPV1. However, the smallest segment of BdPV1/SXD111 differed from that of BdPV1/LW-1. Many mycoviruses have been reported in *B*. *dothidea*. However, no hypovirus has been described in this fungus. Currently, there are eight genera in the family *Hypoviridae*, including *Alphahypovirus*, *Betahypovirus*, *Gammahypovirus*, *Deltahypovirus*, *Epsilonhypovirus*, *Zetahypovirus*, *Etahypovirus*, and *Thetahypovirus* [6]. This study determined and characterized the genome organization of a novel hypovirus BdHV1/SXD111 from *B*. *dothidea*. The genome of BdHV1/SXD111, excluding the poly(A) tail, comprises an ssRNA of 11,128 nt that contains a single putative ORF encoding a large polyprotein. Sequence and phylogenetic analyses of the putative polyprotein strongly support the conclusion that BdHV1/SXD111 is most closely related to members of the genus *Betahypovirus*. Therefore, BdHV1/SXD111 is proposed as a new species in the genus *Betahypovirus* in the family *Hypoviridae*. Furthermore, the most significant difference in genome organization between BdHV1/SXD111 and other hypoviruses reported previously was the length of its 5′-UTR. The 5′-UTR of BdHV1/SXD111 is 913 nt in length and longer than that of the other hypoviruses [5].

Members of the genus *Betahypovirus* have a large ORF that typically encodes a polyprotein, which includes Pro, UGT, RdRp, and Hel conserved domains [5]. These conserved domains were also found in the ployprotein of BdHV1/SXD111. In addition, the protease in the betahypovirus has three conserved Cys, Hys, and Gly residues [41,42,43,44]. In our study, these conserved amino acid residues were identified in the N-terminal region of the polyprotein of BdHV1/SXD111. This result suggests the catalytic site of a highly diverged protease [5, 41–44]. Up to the present day, batahypoviruses have been reported in nine other plant pathogenic fungal species, including *Cryphonectria parasitica* [42,43], *Sclerotinia sclerotiorum* [44,45], *Botrytis cinerea* [46,47], *Fusarium sambucinum* [48], *F*. *oxysporum* [49], *Valsa ceratosperma* [41], *Phomopsis longicolla* [50], *Setosphaeria turcica* [51], *Monilinia fructicola* [52], and *Alternaria dianthicola* [53]. Our study is the first report of a naturally occurring hypovirus that infects the plant pathogenic fungus *B*. *dothidea*. Thus, our findings could expand our knowledge of the diversity of the family *Hypoviridae*.

Interestingly, BdHV1/SXD111 was detected to be co-infected with BdPmV1/SXD111 and BdPV1/SXD111. Coinfection with multiple mycoviruses in plant pathogenic fungi often happens [54]. In *B*. *dothidea* mycovirus-infected strains, an avirulent strain, LW-1, harbored BdCV1 and BdPV1 [19]. In addition, the strain YCLYY11 was coinfected with BdCV1, BdPV1, and BdVV3, belonging to three families [17]. Similar to strain YCLYY11, strain SXD111 was coinfected by three different mycoviruses. These results indicated that BdPmV1, BdCV1, and BdPV1 could spread in natural *B*. *dothidea* populations. Moreover, BdPV1/SXD111 had a novel satellite-like RNA. A noncoding satellite RNA (511 bp in size) has been observed in association with BdPV1/LW-1 [19]. In our study, satellite-like RNA was concurrent with the helper virus in strain SXD111 observed in SC sub-isolates absent of dsRNA1 and dsRNA2 of BdPV1/SXD111 following vertical transmission (data not shown). Additionally, sequence analysis demonstrated that the satellite-like RNA in strain SXD111 lacked a detectable identity with coinfecting dsRNAs, the satellite RNA of BdPV1/LW-1, and the sequences deposited in the NCBI database. Moreover, a conserved heptabase (CGAAAAU) at 5′-termini and a hexabase (ACACCA) at 3′-termini were found in the dsRNA1−3 of BdPV1/SXD111. Similar conserved sequences (CGAAAAU) at 5′-termini and a (CA) at 3′-termini were also present in the genomes of BdPV1/LW-1 [19]. Such conserved sequences of partitiviruses are often thought to be involved in RdRp recognition for RNA packaging and/or replication [55]. Generally, satellite or satellite-like RNA is distinct from its helper virus with a nucleotide sequence with little or no sequence similarity to helper viruses [56]. Therefore, these results indicate that the smallest dsRNA in strain SXD111 could be a noncoding satellite-like RNA and its replication depend on the helper virus BdPV1/SXD111.

Members of the family *Partitiviridae* generally possess two essential dsRNA genome segments [57]. The larger one encodes RdRp, and the smaller one encodes CP on the positive-strand RNA molecule [57]. Moreover, one or more additional dsRNA segments are common in this family. Some of them could even encode a putative protein. For example, Atkinsonella hypoxylon virus [58], Aspergillus ochraceus virus 1 (AoV1) [59], Aspergillus fumigatus partitivirus 1 (AfuPV1) [60], Aspergillus flavus partitivirus 1 (AfPV1) [61], and BdPV2 [20]. By contrast, others could not detect any efficient ORF. For example, BdPV1/LW-1 [19], Ustilaginoidea virens partitivirus 4 (UvPV4) [62], and Penicillium stoloniferum virus F (PsVF) [63]. Interestingly, four dsRNA segments, one with an ORF and another without, were found in partitivirus Discula destructiva virus 1 (DdV1) [64] and the AfPV1/ZD1.22-10-9 which was isolated from the *Aspergillus flavus* strain ZD1.22-10-9 [65]. Even though the RdRp and CP genes of BdPV1/SXD111 had 97.53% and 98.46% homologies with them of BdPV1/LW-1, respectively, their satellite RNA was different. In our previous study, BdVV3, BdCV1, and BdPV1 coinfected a *B*. *dothidea* strain YCLYY11, and the BdPV1/YCLYY11 had two dsRNA segments: only the RdRp and CP genes [17]. Future research is needed to confirm the role of BdPV1/SXD111 and BdPV1/LW-1 satellites.

Many mycoviruses can be transmitted vertically via fungal spores or horizontally via hyphal anastomosis [4]. In this study, only BdHV1/SXD111 could be transmitted to the *B*. *dothidea* strain JNT1111 after dual culture, while BdPmV1/SXD111 and BdPV1/SXD111 could not. Our previous study showed that BdPmV1/YZN115 could be transferred to the strain JNT1111 [23]. Another report showed that BdPV1/LW-1 only could be transmitted to a single conidium strain derived from strain LW-1 but not transferred to different origin strains [19]. Therefore, further studies might explore the factors influencing BdPmV1/SXD111 transmission from strain SXD111 to JNT1111. Moreover, apart from strains SXD111 and YZN115, BdPmV1 also infected other 12 *B*. *dothidea* strains collected from different locations [23]. In our study, the virulence of strain SXD111 was likely at a high level. However, in our previous study, the pathogenicity of 14 BdPmV1-infected strains fluctuated greatly by inducing rotting lesions on pear fruits ranging from 1.5 mm to 38.4 mm in diameter at 5 dpi [23]. Only strains YZN115, EJ2203, SNN221, and LJYa30 showed hypovirulence (rot diameter < 18.0 mm) [23]. Therefore, there was disparate virulence for different strains of BdPmV1.

In our study, the hyphal tip culture, single conidium, and protoplast regeneration method were executed to obtain the virus-free BdHV1/SXD111 isogenic lineage from strain SXD111. However, no virus-free strains were obtained. In the horizontal transmission assay, the BdHV1/SXD111-infected sub-strains SJT1, SJT2, and SJT3 were derived from strain JNT1111. Thus, the strains SJT1, SJT2, SJT3, and JNT1111 of *B*. *dothidea* were compared to evaluate the biological functions of BdHV1/SXD111. However, no apparent morphologic changes, growth rates, or virulence were observed from these strains. These results suggest that the BdHV1/SXD111 likely has no impact on the biological aspects of its host. Notably, the effects of hypovirus infection on their host varied. For example, CHV4/SR2, FgHV1, VcHV1, and Pestalotiopsis fici hypovirus 1 (PfHV1) had no significant effect on their hosts [41,43,66,67]. By contrast, CHV1/EP713, CHV2/NB58, CHV3/GH2, and FgHV2 were associated with the hypovirulence of their host [42,68,69,70,71]. In some hypoviruses, satellite-like RNA is a key for causing the hypovirulence of helper virus. For example, SsHV1/SZ150 coinfection with its satellite-like small dsRNA causes hypovirulence [46]. However, the infection of BcHV1 alone without the satellite RNA significantly reduced virulence without affecting mycelial growth [46]. A recent study showed that satellite-like RNA could negatively regulate the expression of helper virus PfHV1 [66]. Notably, BdHV1/SXD111 had no evident impact on the morphologies and virulence of the host fungus.

## 5. Conclusions

In conclusion, nine dsRNA segments were identified in strain SXD111 isolated from a pear stem wart sample. We discovered a novel hypovirus BdHV1/SXD111, with two known viruses, BdPmV1/SXD111 and BdPV1/SXD111, from strain SXD111 using high-throughput sequencing technology. The genome of BdHV1/SXD111 is 11,128 nucleotides long, excluding the poly (A) tail. Moreover, the Pro, UGT, RdRp, and Hel conserved domains were detected in the polyprotein of BdHV1/SXD111. Phylogenetic analysis showed that BdHV1/SXD111 was clustered with betahypovirus and separated from members of other genera in the family *Hypoviridae*. The BdPmV1/SXD111 genome comprises five dsRNA segments with 2396, 2232, 1967, 1131, and 1060 bp lengths. Additionally, BdPV1/SXD111 harbored three dsRNA segments with 1823, 1623, and 557 bp lengths. Moreover, the smallest dsRNA isolated from strain SXD111 was a novel satellite RNA of BdPV1/SXD111 and different from the satellite RNA of BdPV1/LW-1. BdHV1/SXD111 could be transmitted through conidia and hyphae contact, whereas it likely has no apparent impact on the morphologies and virulence of the host fungus. Many mycoviruses have been reported in *B*. *dothidea*. However, no hypovirus has been described in this fungus. Thus, this study is the first report of a betahypovirus isolated from the fungus *B*. *dothidea*. Importantly, our results significantly enhance the diversity of the *B*. *dothidea* viruses.

## Figures and Tables

**Figure 1 viruses-15-02059-f001:**
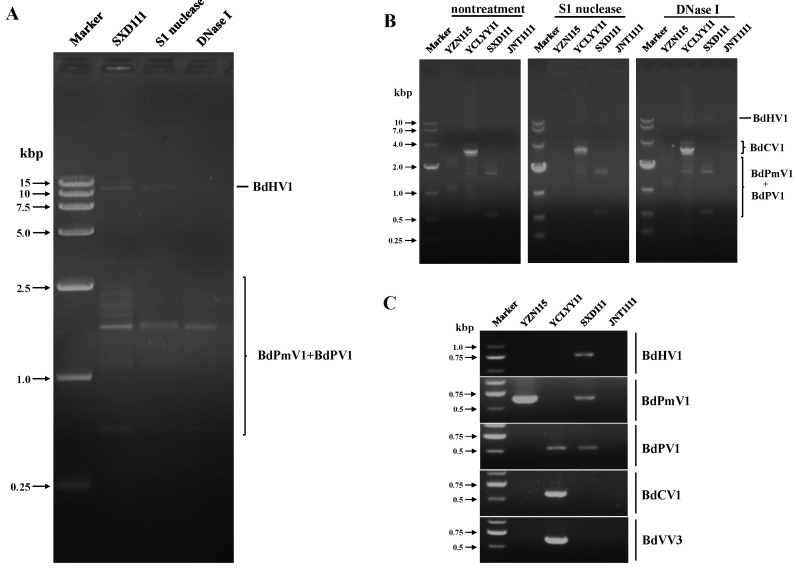
DsRNAs extracted from strain SXD111. (**A**) 1.0% agarose gel electrophoretic profiles of dsRNA preparations extracted from strain SXD111 without treatment (lane 2) and treated with S1 nuclease (lane 3) and DNase I (lane 4), respectively. (**B**) The dsRNA nature of Botryosphaeria dothidea hypovirus 1 (BdHV1), Botryosphaeria dothidea polymycovirus 1 (BdPmV1), Botryosphaeria dothidea partitivirus 1 (BdPV1), Botryosphaeria dothidea chrysovirus 1 (BdCV1), and Botryosphaeria dothidea victorivirus 3 (BdVV3) determined by enzymatic treatment with S1 nuclease and DNase I. (**C**) Detection of BdHV1, BdPmV1, BdPV1, BdCV1, and BdVV3 in the tested strains via RT-PCR using specific primers (Appendix A). The virus-free strain JNT1111 was used as a negative control. Marker, DNA marker.

**Figure 2 viruses-15-02059-f002:**
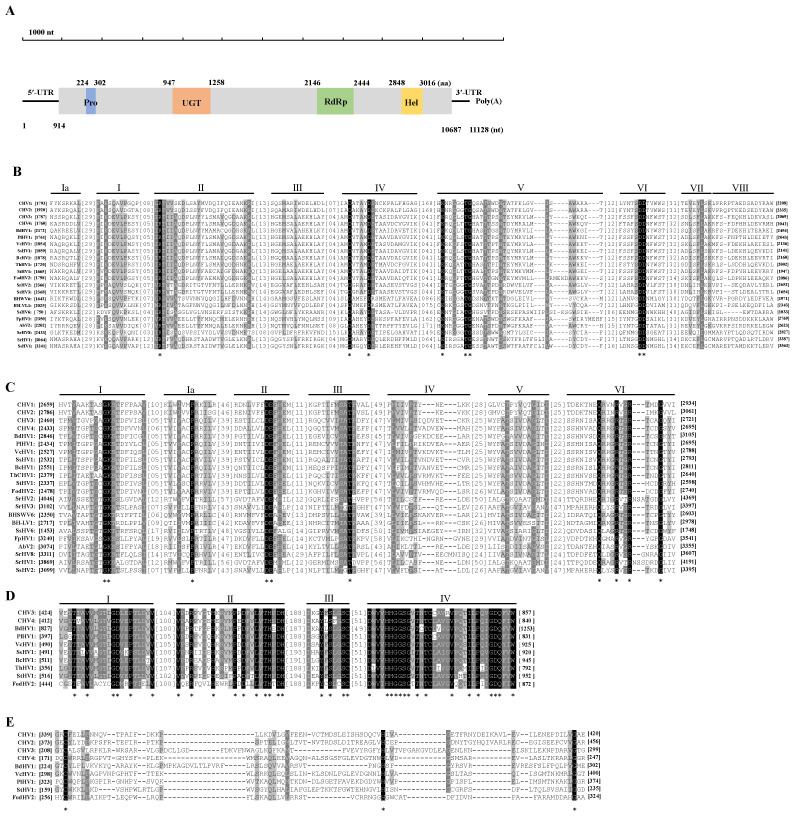
BdHV1/SXD111 genomic organization. (**A**) Genomic organization of the ORF1-encoding polyprotein. The conserved motifs for protease (Pro), UDP-glucose/sterol glucosyltransferase (UGT), RNA-dependent RNA polyprotein (RdRp), and helicase (Hel) domain blocks on the polyprotein are marked in blue, orange, green, and yellow, respectively, with the lengths corresponding to their amino acid (aa) sizes. The numbers under the line indicate the start and end positions of the genome, 5’- and 3’-untranslated regions (UTRs), and the conserved domains. (**B**–**E**) Multiple alignments of the RdRp, Hel, UGT, and Pro of BdHV1/SXD111 amino acid sequences with members of the family *Hypoviridae*. Black shading indicates the conserved sequence level, and the darkest color indicates the most conserved sequence. For abbreviations of viruses and viral protein accession numbers used in alignment analysis, see Appendix A. *, conserved amino acid residues.

**Figure 3 viruses-15-02059-f003:**
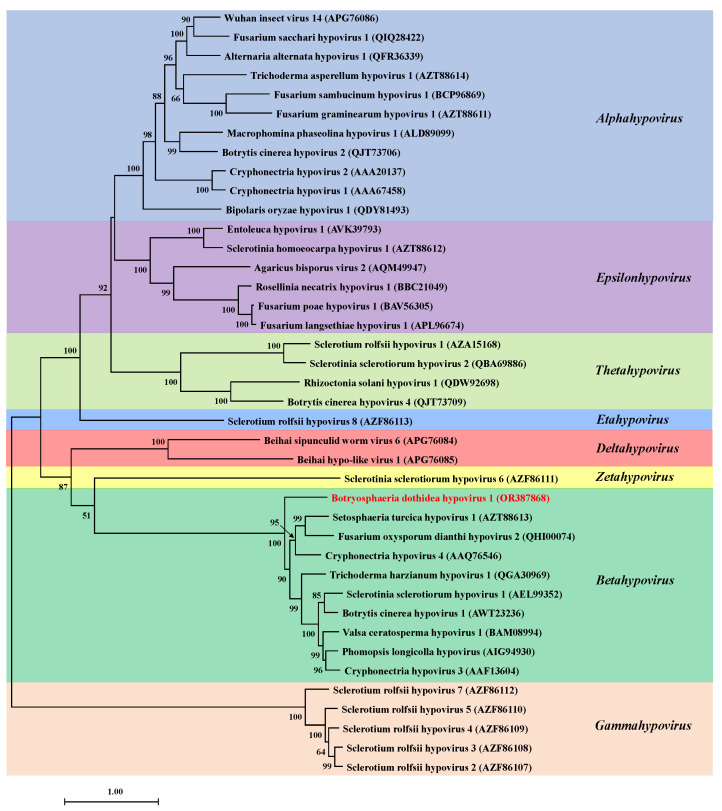
Phylogenetic analysis of Botryosphaeria dothidea hypovirus 1 (BdHV1/SXD111). A maximum-likelihood tree (model LG+G+I+F) was constructed based on the polyprotein sequences encoded by BdHV1/SXD111 and those of known hypoviruses using MEGA X. The putative hypovirus BdHV1/SXD111 is indicated in red. The values in the upper left corner of the branches indicate the bootstrap probability based on 1000 replicates. Bootstrap values < 50% are hidden.

**Figure 4 viruses-15-02059-f004:**
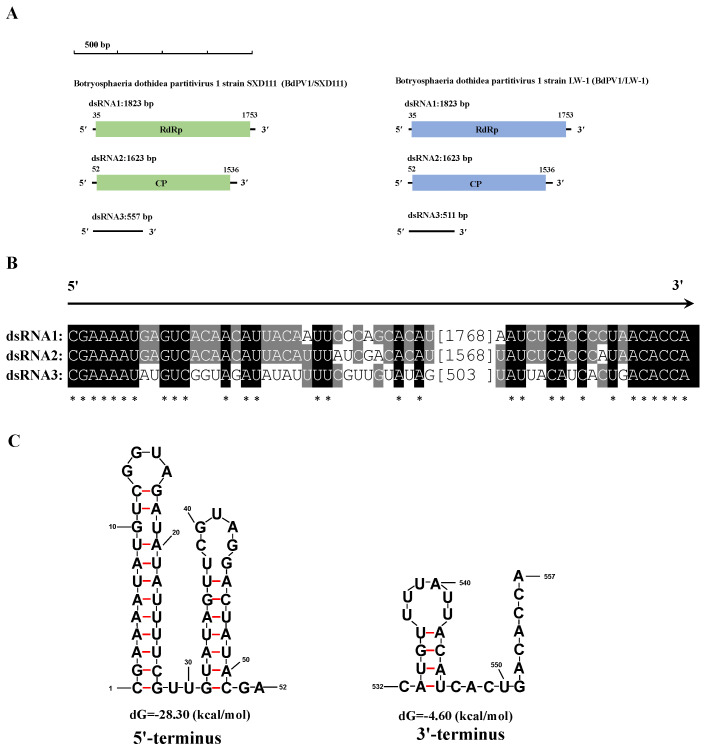
BdPV1/SXD111 genomic organization. (**A**) Schematic representation of the genomic organization of BdPV1/LW-1 and BdPV1/SXD111. The length and sequences of their satellite-like RNA were different. (**B**) Multiple alignment of the dsRNAs of BdPV1/SXD111. Asterisks show identical nucleotides in dsRNAs. (**C**) Secondary structures proposed for 5′- and 3′-terminus of BdPV1/SXD111 dsRNA3 with the lowest energies. *, conserved nucleotides.

**Figure 5 viruses-15-02059-f005:**
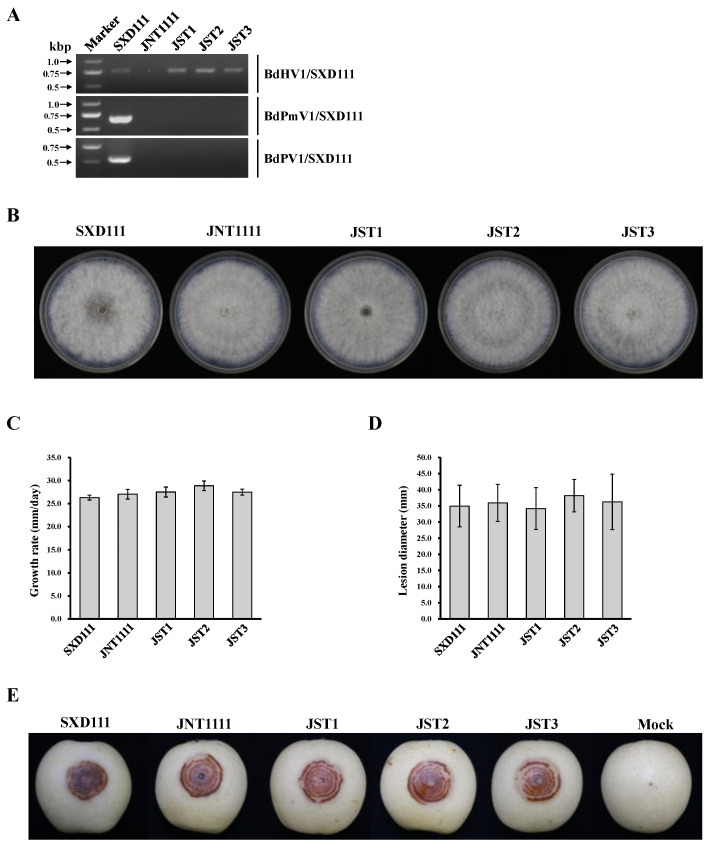
Colony morphology on the PDA plate and virulence on the pear fruit of strains SXD111, JNT1111, JST1, JST2, and JST3. (**A**) Detection of BdHV1/SXD111, BdPmV1/SXD111, and BdPV1//SXD111 in different strains of *B*. *dothidea* via RT-PCR with specific primers (see Appendix A). (**B**) Colony morphology on PDA medium (28 °C, 4 days). (**C**) Statistical analysis of the growth rate of BdHV1/SXD111-infected and virus-free strains on a PDA medium. (**D**) Statistical analysis of the lesion size of BdHV1/SXD111-infected and virus-free strains on wound-inoculated pear (*P*. *bretschneideri* cv. ‘Huangguan’) at four dpi. (**E**) Fruits of pear wound-inoculated with colonized plugs of strains SXD111, JNT1111, JST1, JST2, and JST3 at four dpi. Strains JST1, JST2, and JST3 were derived from JNT1111 in the pairing cultures of strains JNT1111 and SXD111. The error bars indicate the standard deviations from different sample means. Based on a multiple-range test, there was no significant difference at the *p* < 0.05 confidence level for growth rates and lesion sizes.

**Table 1 viruses-15-02059-t001:** Origin of *Botryosphaeria dothidea* strains used in this study.

Strain	Origin	Mycovirus
SXD111	*Pyrus bretschneideri* cv. ‘Hongxiangsu’, Shaanxi province, China	BdHV1 *, BdPmV1, BdPV1
YZN115	*P*. *bretschneideri* cv. ‘Suli’, Henan province, China	BdPmV1
YCLYY11	*Dimocarpus longan* cv. ‘Shuguan’, Chongqing City, China	BdVV3, BdCV1, BdPV1
JNT1111	*P*. *bretschneideri* cv. ‘Suli’, Shanxi, China	Virus free
JST1	JNT1111 in a pairing culture of JNT1111 and SXD111	BdHV1
JST2	JNT1111 in a pairing culture of JNT1111 and SXD111	BdHV1
JST3	JNT1111 in a pairing culture of JNT1111 and SXD111	BdHV1

* Abbreviations of mycoviruses: Botryosphaeria dothidea hypovirus 1 (BdHV1), Botryosphaeria dothidea polymycovirus 1 (BdPmV1), Botryosphaeria dothidea partitivirus 1 (BdPV1), Botryosphaeria dothidea chrysovirus 1 (BdCV1), and Botryosphaeria dothidea victorivirus 3 (BdVV3).

**Table 2 viruses-15-02059-t002:** Best BLASTx matches of contigs obtained in this study.

No.	Contig Number	Contig Length (nt/bp)	Best Match	Protein	Cover %	Aa Ident %	Taxon
1	contig 1	11,113	Cryphonectria hypovirus 4 (CHV4)	polyprotein	68	49.77	*Hypoviridae* *Betahypovirus*
2	contig 135	3592	Botryosphaeria dothidea chrysovirus 1 (BdCV1)	RdRp	93	98.66	*Chrysoviridae* *Betachrysovirus*
3	contig 2025	1401		hypothetical protein P3	99	99.14	
4	Contig 2640	1216		coat protein	85	98.90	
6	contig 5902	703		hypothetical protein	91	98.13	
7	contig 744	2171	Botryosphaeria dothidea polymycovirus 1 (BdPmV1)	hypothetical protein P2	95	100.00	*Polymycoviridae Polymycovirus*
8	contig 1102	1855		methyltransferase	97	99.17	
9	contig 3852	960		RdRp	99	99.37	
10	contig 5123	784		hypothetical protein P5	89	98.29	
11	Contig 5986	694		PAS-rich protein	97	98.22	
12	contig 1458	1635	Botryosphaeria dothidea partitivirus 1 (BdPV1)	coat protein	90	99.80	*Partitiviridae*
13	contig 5134	783		RdRp	97	100	
14	contig 406	2641	Botryosphaeria dothidea victorivirus 3 (BdVV3)	RdRp	93	99.80	*Totiviridae Victorivirus*
15	contig 421	2624		coat protein	81	99.86	

## Data Availability

Not applicable.

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
