# Peer review of "Identification and Characterization of a Novel Hypovirus from the Phytopathogenic Fungus Botryosphaeria dothidea"

_viruses, 2023, doi:10.3390/v15102059_

Round 1

Reviewer 1 Report

The authors present some interesting work on mycoviruses in Botryosphaeria dothidia.  There is no question that the research that has been done adds to the field of mycoviruses in general and specifically to the genus Botryosphaeria.

I found that that the manuscript was very confusing and often contradictory making it very difficult to understand some of what the authors are presenting.  The grammar and use of English also needs some attention.

FOR EXAMPLE LANGUAGE PROBLEMS:

“Recent studies have also  demonstrated that nine positive ssRNA viruses infected this fungus have been reported,  including Botryosphaeria dothidea fusarivirus 1 (BdFV1) and BdFV2 belongs to the family Fusariviridae”

Recent studies have also demonstrated that nine positive ssRNA viruses infectING this fungus have been reported, including Botryosphaeria dothidea fusarivirus 1 (BdFV1) and BdFV2 WHICH belongs to the family Fusariviridae

Groupes  should be groups

Moreso – this is not a word?

EXAMPLES OF CONFUSING STATEMENTS:

Thus, this  study is the first report of a hypovirus isolated from the fungus Bdothidea and contributes useful information to better understand viral diversity in this fungus – BUT NO COMMENT ABOUT THE OTHER VIRUSES?

In abstract it is stated “This study identified a range of dsRNA segments in B.dothidea strain SXD111 isolated from a pear stem wart sample” BUT IN THE MATERIALS AND METHODS 10 STRAINS WERE USED?

Multiple sequence alignment based on the amino acid sequence of the RdRp domains  of BdHV1/SXD111 and other hypoviruses revealed nine conserved motifs in these hypoviruses (Figure 2B). 9 DOMAINS IN ALL VIRUSES OR EACH VIRUS – DOES NOT MAKE SENSE.

The manuscript would benefit from a table summarising the different viruses that have been isolated from these strains.

In the discussion section it is stated:

“To date, a few Bdothidea strains” and then in the next paragraph there is the comment “Viruses are commonly present in Bdothidea”.  These two statements are contradictory

Overall ,I suggest that the author, greatly improve the logic and flow of the ideas in the manuscript.  The abstract in particular does not reflect the research that was done.  The last paragraph in the Introduction is out of context.  The authors should consider having the same subtitles in their materials and results section.  This would allow the reader to see firstly what was done and then what the results were from the experiments in a logical order.  At present the flow is not obvious and it makes reading the manuscript very difficult.

I also note it is not stated what sequencing methodology was used?

The grammar and use of English also needs some attention.

Author Response

Review 1

The authors present some interesting work on mycoviruses in Botryosphaeria dothidea.  There is no question that the research that has been done adds to the field of mycoviruses in general and specifically to the genus Botryosphaeria.

I found that that the manuscript was very confusing and often contradictory making it very difficult to understand some of what the authors are presenting. The grammar and use of English also needs some attention.

Response: Thank you for your comments concerning our manuscript (Manuscript ID: viruses-2596773). Those comments are all valuable and very helpful for revising and improving the quality of our manuscript. We have checked the manuscript carefully and selected the English editing service provided by AiMi Academic Services (www.aimieditor.com). Revised portions are marked in green in the revised manuscript.

FOR EXAMPLE LANGUAGE PROBLEMS:

“Recent studies have also demonstrated that nine positive ssRNA viruses infected this fungus have been reported, including Botryosphaeria dothidea fusarivirus 1 (BdFV1) and BdFV2 belongs to the family Fusariviridae”

Recent studies have also demonstrated that nine positive ssRNA viruses infectING this fungus have been reported, including Botryosphaeria dothidea fusarivirus 1 (BdFV1) and BdFV2 WHICH belongs to the family Fusariviridae

Response: the sentences was replaced in lines 52-54 in the revised manuscript.

Groupes should be groups

Response: we changed “Groupes” to “Groups” in line 57 in the revised manuscript.

Moreso – this is not a word?

Response: the word “moreso” was deleted in the revised manuscript.

EXAMPLES OF CONFUSING STATEMENTS:

Thus, this study is the first report of a hypovirus isolated from the fungus B. dothidea and contributes useful information to better understand viral diversity in this fungus – BUT NO COMMENT ABOUT THE OTHER VIRUSES?

Response: the sentence was replaced by “Thus, this study is the first report of a betahypovirus isolated from the fungus B. dothidea. Importantly, our results significantly enhanced the diversity of the B. dothidea viruses.” in lines 28-29 in the revised manuscript.

In abstract it is stated “This study identified a range of dsRNA segments in B.dothidea strain SXD111 isolated from a pear stem wart sample” BUT IN THE MATERIALS AND METHODS 10 STRAINS WERE USED?

Response: To verify mycovirus in strain SXD111 and the impact of BdHV1 on the morphologies and virulence of the host fungus. Seven strains of B. dothidea were used in this study. Strains YZN115 and YCLYY11 were used as positive control in high-throughput sequencing because strains YZN115 and YCLYY11 were infected by known mycoviruses. Strains JST1, JST2, and JST3 isolated from strain JNT1111 were obtained from the horizontal transmission assay. The virus-free strain JNT1111 was used as a control for evaluating biological characteristics.

Multiple sequence alignment based on the amino acid sequence of the RdRp domains of BdHV1/SXD111 and other hypoviruses revealed nine conserved motifs in these hypoviruses (Figure 2B). 9 DOMAINS IN ALL VIRUSES OR EACH VIRUS – DOES NOT MAKE SENSE.

Response: This sentence was replaced by “Multiple sequence alignment based on the amino acid sequence of the RdRp do-mains of BdHV1/SXD111 and other hypoviruses revealed nine conserved motifs in these hypoviruses, including the S/GDD motif, which is a typical feature in RdRp of hypoviruses.” in lines 263-266 in the revised manuscript.

In other hand, we labeled the motifs in the Figure 2 in the revised manuscript.

The manuscript would benefit from a table summarising the different viruses that have been isolated from these strains.

Response: Thank you for your advice. The different viruses that have been isolated from these strains were showed in the table 1.

In the discussion section it is stated:

“To date, a few B. dothidea strains” and then in the next paragraph there is the comment “Viruses are commonly present in B. dothidea”. These two statements are contradictory.

Response: Thank you for your advice. the sentence “a few B. dothidea strains” was replaced by “many B. dothidea strains” in the revised manuscript.

Overall, I suggest that the author, greatly improve the logic and flow of the ideas in the manuscript. The abstract in particular does not reflect the research that was done. The last paragraph in the Introduction is out of context. The authors should consider having the same subtitles in their materials and results section. This would allow the reader to see firstly what was done and then what the results were from the experiments in a logical order. At present the flow is not obvious and it makes reading the manuscript very difficult.

Response: Thank you for your advice. We have checked the manuscript carefully and revised our manuscript. Revised portions are marked in green in the revised manuscript.

I also note it is not stated what sequencing methodology was used?

Response: The sequencing methodology was described in lines 106-114 in the revised manuscript.

Reviewer 2 Report

This manuscript describe the molecular characterization of three micoviruses  isolated from Botryosphaeria dothidea strain SXD111. One of them correspond to an unknown virus that according to its  genomic structure,  would be a new member of the genus Betahypovirus of the family Hypoviridae. It was tentatively named Botryosphaeria dothidea hypovirus 1 (BdHV1/SXD111). The genomic sequences of the others virus were also  characterized. These were named named Botryosphaeria dothidea polymycovirus 1 strain SXD111 (BdPmV1/SXD111) and Botryosphaeria dothidea partitivirus 1 strain SXD111 (BdPV1/SXD111). Besides, the manuscript reported the horizontal and vertical transmission of these viruses, and the biological effect of BdHV1 on  B. dothidea.

In general, the introduction provides sufficient background and includes all relevant references. The methods about molecular characterization of the mycovirus are adequately described and the results are clearly presented. The results are  interesting because  report the whole sequences of these three viruses. It include the first report of  a virus (BdHV1)  belonging to the family Hypoviridae in this fungal species, and the analysis of a satellite virus in  BdPV1 that deserves to be studied in depth. The results on horizontal transmission of viruses raise some interesting questions that deserve to be highlighted.

Comments:

1) There are many typographical and grammatical errors throughout the manuscript. The manuscript could be improved by having someone with English as their native language read it.

2) Line 71: I did not find studies on Botryosphaeria dothidea strain SXD111 at the indicated citation. Please revise.

3) Line 78: I understand that the BdRV1/SXD111 virus is then indicated as BdPmV1/SXD111. I think this is confusing throughout the manuscript and should be better presented

4) Line 164: Please, indicate the method to evaluate the viral presence in this experiment (RT-PCR).

5) Line 167-171: The horizontal transmission experiment did not include a selective isolation of the recipient strain. I understand that the isolation of the receiving strain is carried out far from the contact zone. If they used any method to make sure to isolate the recipient strain, they should indicate it. If not, I suggest that you add a schematic figure that shows the isolation area.

6) Line 315: Same as Line 78.

7) Line 357: Throughout the manuscript there are several sentences that should improve the writing. Please check the sentence of line 357

8) Line 361:  The authors did no report any attenuation effect of the naturally infected strain (SXD 111) or the artificially infected strain (JST1-3) respect to the virus-free strain (JNT1111). Interestingly, the SXD111 strain presents a virus (BdPmV1) that has been reported to have effects on its host (YZN115).  I think it would be very convenient to include strain YZN115 as a control in this experiment and compare the accumulation levels of each virus in all strains (SXD 111,  YZN115 and JST1-3).

9) Line 395:  please revise.

10) Line 442: BdRV1 (BdPmV1 ) or BdVP1?. Please revise

11) Lines 473-484:  I think that very interesting topics are addressed here that could be more discussed. Why does the BdPmV1 virus not exert effects? Is there antagonism with the rest of the viruses? Why are BdPmV1 and BdPV1 not transmitted horizontally? How is this behavior compared to the antecedents in these viruses? could the satellite virus have an effect? Is there anything to say about the viral accumulation of these viruses in the strain studied?

.

Author Response

Review 2

This manuscript describe the molecular characterization of three micoviruses isolated from Botryosphaeria dothidea strain SXD111. One of them correspond to an unknown virus that according to its genomic structure, would be a new member of the genus Betahypovirus of the family Hypoviridae. It was tentatively named Botryosphaeria dothidea hypovirus 1 (BdHV1/SXD111). The genomic sequences of the others virus were also characterized. These were named named Botryosphaeria dothidea polymycovirus 1 strain SXD111 (BdPmV1/SXD111) and Botryosphaeria dothidea partitivirus 1 strain SXD111 (BdPV1/SXD111). Besides, the manuscript reported the horizontal and vertical transmission of these viruses, and the biological effect of BdHV1 on B. dothidea.

In general, the introduction provides sufficient background and includes all relevant references. The methods about molecular characterization of the mycovirus are adequately described and the results are clearly presented. The results are interesting because report the whole sequences of these three viruses. It include the first report of a virus (BdHV1) belonging to the family Hypoviridae in this fungal species, and the analysis of a satellite virus in BdPV1 that deserves to be studied in depth. The results on horizontal transmission of viruses raise some interesting questions that deserve to be highlighted.

Comments:

1) There are many typographical and grammatical errors throughout the manuscript. The manuscript could be improved by having someone with English as their native language read it.

Response: Thank you for your advice. We have revised the manuscript carefully and selected the English editing service provided English language by the AiMi Academic Services (www.aimieditor.com). Revised portions are marked in green in the revised manuscript.

2) Line 71: I did not find studies on Botryosphaeria dothidea strain SXD111 at the indicated citation. Please revise.

Response: Thank you for your advice. We redescribed the origin of strain SXD111 in lines 82-84 in the revised manuscript.

3) Line 78: I understand that the BdRV1/SXD111 virus is then indicated as BdPmV1/SXD111. I think this is confusing throughout the manuscript and should be better presented.

Response: Yes, the BdRV1/SXD111 virus was indicated as BdPmV1/SXD111 in the revised manuscript.

4) Line 164: Please, indicate the method to evaluate the viral presence in this experiment (RT-PCR).

Response: Thank you for your advice. the RT-PCR method to evaluate the viral presence were added in lines 176-177 in the revised manuscript.

5) Line 167-171: The horizontal transmission experiment did not include a selective isolation of the recipient strain. I understand that the isolation of the receiving strain is carried out far from the contact zone. If they used any method to make sure to isolate the recipient strain, they should indicate it. If not, I suggest that you add a schematic figure that shows the isolation area.

Response: Thank you for your advice. In our horizontal transmission experiment, the derivative strains were obtained from the colonial margin of strain JNT1111 (the farthest site from the contact zone of strains SXD111 and JNT1111). We added the sentence in lines 174-176 in the revised manuscript.

6) Line 315: Same as Line 78.

Response: the “BdRV1/SXD111” was replaced by “BdPmV1/SXD111”.

7) Line 357: Throughout the manuscript there are several sentences that should improve the writing. Please check the sentence of line 357

Response: the sentence was replaced by “The result of RT-PCR for detecting BdHV1/SXD111, BdPmV1/SXD111, and BdPV1/SXD111 demonstrated that sub-strains SJT1, SJT2 and SJT3 harbored BdHV1/SXD111. However, BdPmV1/SXD111 and BdPV1/SXD111 were negatively detected in all sub-strains SJT1, SJT2 and SJT3.” in lines 360-363 in the revised manuscript. And some others were improved the writing. Revised portions are marked in green in the revised manuscript.

8) Line 361: The authors did no report any attenuation effect of the naturally infected strain (SXD111) or the artificially infected strain (JST1-3) respect to the virus-free strain (JNT1111). Interestingly, the SXD111 strain presents a virus (BdPmV1) that has been reported to have effects on its host (YZN115). I think it would be very convenient to include strain YZN115 as a control in this experiment and compare the accumulation levels of each virus in all strains (SXD111, YZN115 and JST1-3).

Response: In this study, our purpose is the effect of BdHV1, therefor, strain YZN115 was not used in this experiment. However, our previous study showed that the pathogenicity of 14 BdPmV1-infected strains fluctuated greatly by inducing rotting lesions on the pear fruits ranging from 1.5 mm to 38.4 mm in diameters at 5 dpi (Zhai et al., 2016. Supplemental Table S3). In that study, the virulence of strain SXD111 was also high level (rot diameter=34.0 mm but strain YZN115=12.2 mm).

9) Line 395:  please revise.

Response: Thanks, this sentence was replaced by “Viruses are commonly present in B. dothidea. Currently, 18 mycoviruses in 10 families have been isolated from this fungus.” In lines 399-400 in the revised manuscript.

10) Line 442: BdRV1 (BdPmV1) or BdVP1?. Please revise

Response: It is BdPV1, we revised it in lines 444 in the revised manuscript.

11) Lines 473-484: I think that very interesting topics are addressed here that could be more discussed. Why does the BdPmV1 virus not exert effects? Is there antagonism with the rest of the viruses? Why are BdPmV1 and BdPV1 not transmitted horizontally? How is this behavior compared to the antecedents in these viruses? could the satellite virus have an effect? Is there anything to say about the viral accumulation of these viruses in the strain studied?

Response: Thank you for your advice. These questions are also that we are confused. Certainly, the pathogenicity of BdPmV1-infected strains was fluctuated greatly (Zhai et al., 2016. Supplemental Table S3). It might more sequence for BdPmV1 to analyze the evolutionary. In our study, the satellite-like RNA was concurrent with the helper virus in strain SXD111 observed in any sub-isolates absent of dsRNA1 and dsRNA2 of BdPV1/SXD111 following vertical transmission. Therefore, the effect of satellite virus was undefined. I think that your advice points out the direction for our next research.

Reviewer 3 Report

The study profiled dsRNA segments and determined the sequences of each. Among the viruses, one hypovirus was a novel virus discovered in this study. The study also compared the pathogenicity of several fungal isolates whether there were obvious effects on the morphologies and virulence of B. dothidea and found no difference. They attempted hyphal tip, single conidium, and protoplast regeneration but all failed (in the authors' own words). The authors could have tried to clone the virus and make in vitro transcripts for transformation of the fungal protoplasts. But this reviewer see the merit in the study that the authors characterized all the sequences of the bands shown on the dsRNA electrophoresis gel. The authors therefore are suggested to tone down all the claims on the morphologic changes, growth rates or virulence from the strains. Add "likely" to all those sentences. For example, Line 492, likely has no impact on etc.

I suggest to delete table 3 or move it to supplemental because the sequence is publicly available and as the database is changing everyday with more viruses discovered, this blastX result can change over time.

Some English editing is required, such as Line 21 starting as "Other," and Line 227 is better to start as "dsRNA bands similar in size".

Author Response

Review 3

The study profiled dsRNA segments and determined the sequences of each. Among the viruses, one hypovirus was a novel virus discovered in this study. The study also compared the pathogenicity of several fungal isolates whether there were obvious effects on the morphologies and virulence of B. dothidea and found no difference. They attempted hyphal tip, single conidium, and protoplast regeneration but all failed (in the authors' own words). The authors could have tried to clone the virus and make in vitro transcripts for transformation of the fungal protoplasts. But this reviewer see the merit in the study that the authors characterized all the sequences of the bands shown on the dsRNA electrophoresis gel. The authors therefore are suggested to tone down all the claims on the morphologic changes, growth rates or virulence from the strains. Add "likely" to all those sentences. For example, Line 492, likely has no impact on etc.

Response: Thank you for your comments concerning our manuscriptThose comments are all valuable and very helpful for revising and improving the quality of our manuscript, as well as the important guiding significance to our researches.

I suggest to delete table 3 or move it to supplemental because the sequence is publicly available and as the database is changing everyday with more viruses discovered, this blastX result can change over time.

Response: Thank you for your advice, the table 3 was moved to supplemental Table 2 in the revised manuscript.

Some English editing is required, such as Line 21 starting as "Other," and Line 227 is better to start as "dsRNA bands similar in size".

Response: We have revised the manuscript carefully and selected the English editing service provided English language by the AiMi Academic Services (www.aimieditor.com). Revised portions are marked in green in the revised manuscript.

Round 2

Reviewer 3 Report

I saw some minor typos, such as primer not prime. The authors should go over their own manuscript again carefully.

Same as above.

Author Response

Comments and Suggestions for Authors

I saw some minor typos, such as primer not prime. The authors should go over their own manuscript again carefully.

Response: Thank you for your comments concerning our manuscript. Those comments are all valuable and very helpful for revising and improving the quality of our manuscript. The main corrections in the paper and the responses to the comments are as flowing:

Abstract:

  1. We deleted the sentence “Botryosphaeria dothidea, a widespread plant pathogenic fungus, can cause leaf spot, stem canker, and fruit rot diseases on many economically important plants worldwide.”
  2. changed “Many mycoviruses have been accurately and successfully identified in this fungus.” to “Many mycoviruses have been accurately and successfully identified in plant pathogenic fungus Botryosphaeria dothidea” in lines 12-13 in the revised manuscript.

Introduction:

  1. added the “stem wart” symptom in line 45 in the revised manuscript.
  2. changed “This family's 7.3–18.3 kb genomes encode either a single large ORF or two ORFs” to “The genomes of this family encode either a single large ORF or two ORFs” in line 38 in the revised manuscript.
  3. deleted the “B. dothidea.” in line 48 in the revised manuscript.
  4. changed “in B. dothidea strains” to “isolated from B. dothidea” in line 64 in the revised manuscript.
  5. changed “In this study, three mycoviruses, a novel hypovirus, BdPmV1 and Botryosphaeria dothidea partitivirus 1 (BdPV1), were detected from the strain SXD111 using high-throughput sequencing technology and reverse-transcription polymerase chain reaction (RT-PCR).” to “In this study, three mycoviruses, a novel hypovirus and two known mycoviruses, were detected from a strain SXD111 of B. dothidea.” in lines 68-69 in the revised manuscript.

Materials and Methods:

  1. changed “Strain SXD111 of B. dothidea were isolated from a stem scar on a pear tree.” to “Strain SXD111 was isolated from a stem wart sample on a pear tree” in lines 78-79 in the revised manuscript.
  2. changed “Mycovirus” to “Mycoviruses” in Table 1.
  3. changed “5” to “five” in line 104 in the revised manuscript.
  4. changed “Nanodrop” to “a Nanodrop” and “qualified” to “the qualified” in line 107 in the revised manuscript.
  5. changed “prime” to “primer” in line 112 in the revised manuscript.
  6. changed “Mycelia of strains SXD111, YZN115, YCLYY11, and JNT1111 were separately cul-tured on cellophane membranes laid on PDA plates (9 cm in diameter) for 5 days at 28°C. Fresh mycelia were harvested and ground into a fine powder in liquid nitrogen. The extraction of dsRNA was performed using the cellulose chromatography (Sigma–Aldrich, Dorset, England) method.” to “The extraction of dsRNA from strains SXD111, YZN115, YCLYY11, and JNT1111 was performed using the cellulose chromatography (Sigma–Aldrich, Dorset, England) method.” in lines 119-120 in the revised manuscript.
  7. changed “We used RT-PCR to determine mycoviruses in the tested strains using specific primers designed based on the assembled contigs.” to “We used reverse-transcription polymerase chain reaction (RT-PCR) to determine mycoviruses in the tested strains using specific primers designed according to the assembled contigs.” in lines 126-127 in the revised manuscript.
  8. changed “sequences” to “sequence” and “were” to “was” in line 133 in the revised manuscript.
  9. changed “phylogenetic trees were constructed” to “phylogenetic tree was constructed” in line 161 in the revised manuscript.
  10. changed “Bootstrap values were evaluated” to “Bootstrap value was evaluated” in line 162 in the revised manuscript.
  11. changed “Furthermore, the presence of virus in each mycelial derivative strain was determined by RT-PCR using specific primers” to “Furthermore, the presence of viruses in derivative strains SJT1, SJT2 and SJT3 were determined by RT-PCR using specific primers” in lines 175-176 in the revised manuscript.
  12. changed “Moreover, colony morphology, growth rates, and virulence of the derivatives were evaluated as described above and compared with those of their parental strains SXD111 and JNT1111.” to “Moreover, colony morphology, growth rates, and virulence of the derivative strains were evaluated and compared with those of their parental strains SXD111 and JNT1111.” in lines 176-178 in the revised manuscript.
  13. changed “diameter, 5 mm” to “5 mm in diameter” in line 181 in the revised manuscript.
  14. changed “90 mm diameter” to “90 mm in diameter” in line 182 in the revised manuscript.
  15. changed “inoculations” to “inoculation” in line 190 in the revised manuscript.

Results:

  1. changed “To identify the presence of mycoviruses in the fungi of B. dothidea strain SXD111” to “To identify the presence of mycoviruses in the strain SXD111 of B. dothidea” in line 195 in the revised manuscript.
  2. changed “Detection of BdHV1, BdPmV1, BdPV1, BdCV1, and BdVV3 in them host strains YCLYY11 by RT-PCR using specific primers” to “Detection of BdHV1, BdPmV1, BdPV1, BdCV1, and BdVV3 in the tested strains by RT-PCR using specific primers” in lines 234-236 in the revised manuscript.
  3. changed “it” to “the protein” in line 250 in the revised manuscript.
  4. changed “identities” to “identity” in lines 251 and 252 in the revised manuscript.
  5. changed “alignments” to “alignment” in lines 263 and 266 in the revised manuscript.
  6. after “papain-like cysteine protease” added “(Pro)” in line 276 in the revised manuscript.
  7. changed “are” to “were” in line 307 in the revised manuscript.
  8. changed “were 100% identical” to “were identical” in line 333 in the revised manuscript.
  9. changed “was” to “were” in line 334 in the revised manuscript.
  10. changed “alignment” to “alignment” in line 343 in the revised manuscript.
  11. changed “was undetected” to “was negatively detected” in line 350 in the revised manuscript.
  12. changed “could have vertical transmission in its host strain” to “could vertically transmit in host strain” in line 351 in the revised manuscript.
  13. deleted “SJT1, SJT2 and SJT3” in line 356 in the revised manuscript.
  14. changed “utilized” to “determined” in line 362 in the revised manuscript.
  15. changed “Colony morphology and virulence on the pear fruit of strains SXD111 and JNT1111 and three derivate sub-strains JST1−3.” to “Colony morphology on the PDA plate and virulence on the pear fruit of strains SXD111, JNT1111, JST1, JST2 and JST3.” in line 374-375 in the revised manuscript.
  16. changed “4” to “four” in lines 378 and 382 in the revised manuscript.
  17. changed “strain” to “strains” in line 383 in the revised manuscript.
  18. changed “the growth rate and lesion size” to “the growth rates and lesion sizes” in line 386 in the revised manuscript.

Discussion:

  1. changed “In this study, from a B. dothidea strain SXD111, we isolated and characterized a novel hypovirus, BdPmV1, and BdPV1 with a novel satellite-like RNA.” to “In this study, from a B. dothidea strain SXD111, we isolated and characterized a novel hypovirus coinfected with BdPmV1/SXD111 and BdPV1/SXD111. Additionally, BdPV1/SXD111 harbored a novel satellite-like RNA.” in lines 389-392 in the revised manuscript.
  2. changed “strains” to “strain” in line 397 in the revised manuscript.
  3. changed “This study determined and characterized the genome organization of BdHV1/SXD111. The genome of BdHV1/SXD111, excluding the poly(A) tail, comprises an ssRNA of 11128 nt that contains a single putative ORF encoding a large polyprotein.” to “This study determined and characterized the genome organization of a novel hypovirus BdHV1/SXD111. The genome of BdHV1/SXD111, excluding the poly(A) tail, comprises an ssRNA of 11128 nt that contains a single putative ORF encoding a large polyprotein.” in lines 406-409 in the revised manuscript.
  4. changed “Members of the genus Betahypovirus have a large ORF that typically encodes a polyprotein which includes Pro, UGT, RdRp, and Hel. In addition, the protease in betahypovirus has three conserved Cys, Hys, and Gly residues.” to “Members of the genus Betahypovirus have a large ORF that typically encodes a polyprotein which includes Pro, UGT, RdRp, and Hel conserved domains. These conserved domains were also found in the polyprotein of BdHV1/SXD111.” In lines 416-418 in the revised manuscript.
  5. changed “indicate” to “indicated” in line 434 in the revised manuscript.
  6. changed “A satellite RNA has been observed in association with BdPV1/LW-1, dsRNA3 (511 bp in size) of BdPV1/LW-1 in B. dothidea strain LW-1, which encoded no protein.” to “A noncoding satellite RNA (511 bp in size) has been observed in association with BdPV1/LW-1.” in lines 436-437 in the revised manuscript.
  7. changed “any sub-isolates” to “SC sub-strains” in line 438 in the revised manuscript.
  8. deleted “a” before “hexabase” in line 443 in the revised manuscript.
  9. changed “satellite” to “satellite-like” in line 450 in the revised manuscript.
  10. added “coat protein” before “CP” in line 453 in the revised manuscript.
  11. changed “Interestingly, four dsRNA segments, one with an ORF and another without, were found in a partitivirus Discula destructiva virus 1 (DdV1) and AfPV1 strain ZD1.22-10-9 which was isolated in Aspergillus flavus strain ZD1.22-10-9.” to “Interestingly, four dsRNA segments, one with an ORF and another without, were found in Discula destructiva virus 1 (DdV1) and AfPV1/ZD1.22-10-9 which was isolated from Aspergillus flavus strain ZD1.22-10-9.” in lines 460-462 in the revised manuscript.
  12. added “other” before “12” in line 476 in the revised manuscript.
  13. changed “We discovered a novel hypovirus BdHV1/SXD111, two were known viruses BdPmV1/SXD111 and BdPV1/SXD111 from strain SXD111 using high-throughput sequencing technology.” to “We discovered a novel hypovirus BdHV1/SXD111, two known viruses BdPmV1/SXD111 and BdPV1/SXD111 from strain SXD111.” in lines 503-504 in the revised manuscript.
  14. changed “Moreover, a papain-like cysteine protease (Pro), a UDP-glucose/sterol glucosyltrans-ferase (UGT), an RNA-dependent RNA polyprotein (RdRp), and a helicase (Hel) were detected in the polyprotein of BdHV1/SXD111.” to “Moreover, the Pro, UGT, RdRp, and Hel conserved domains were detected in the polyprotein of BdHV1/SXD111.” in lines 505-507 in the revised manuscript.

References

  1. deleted reference 72.
